# A simple example of Dirichlet process mixture inconsistency for the number of components

**Jeffrey W. Miller**
Division of Applied Mathematics
Brown University
Providence, RI 02912
jeffrey_miller@brown.edu

**Matthew T. Harrison**
Division of Applied Mathematics
Brown University
Providence, RI 02912
matthew_harrison@brown.edu

## Abstract

For data assumed to come from a finite mixture with an unknown number of components, it has become common to use Dirichlet process mixtures (DPMs) not only for density estimation, but also for inferences about the number of components. The typical approach is to use the posterior distribution on the number of clusters — that is, the posterior on the number of components represented in the observed data. However, it turns out that this posterior is not consistent — it does not concentrate at the true number of components. In this note, we give an elementary proof of this inconsistency in what is perhaps the simplest possible setting: a DPM with normal components of unit variance, applied to data from a "mixture" with one standard normal component. Further, we show that this example exhibits severe inconsistency: instead of going to 1, the posterior probability that there is one cluster converges (in probability) to 0.

## 1 Introduction

It is well-known that Dirichlet process mixtures (DPMs) of normals are consistent for the density — that is, given data from a sufficiently regular density $p_0$ the posterior converges to the point mass at $p_0$ (see [1] for details and references). However, it is easy to see that this does not necessarily imply consistency for the number of components, since for example, a good estimate of the density might include superfluous components having vanishingly small weight.

Despite the fact that a DPM has infinitely many components with probability 1, it has become common to apply DPMs to data assumed to come from finitely many components or "populations", and to apply the posterior on the number of clusters (in other words, the number of components used in the process of generating the observed data) for inferences about the true number of components; see [2, 3, 4, 5, 6] for a few prominent examples. Of course, if the data-generating process very closely resembles the DPM model, then it is fine to use this posterior for inferences about the number of clusters (but beware of misspecification; see Section 2). However, in the examples cited, the authors evaluated the performance of their methods on data simulated from a fixed finite number of components or populations, suggesting that they found this to be more realistic than a DPM for their applications.

Therefore, it is important to understand the behavior of this posterior when the data comes from a finite mixture — in particular, does it concentrate at the true number of components? In this note, we give a simple example in which a DPM is applied to data from a finite mixture and the posterior distribution on the number of clusters does not concentrate at the true number of components. In fact, DPMs exhibit this type of inconsistency under very general conditions [7] — however, the aim of this note is brevity and clarity. To that end, we focus our attention on a special case that is as

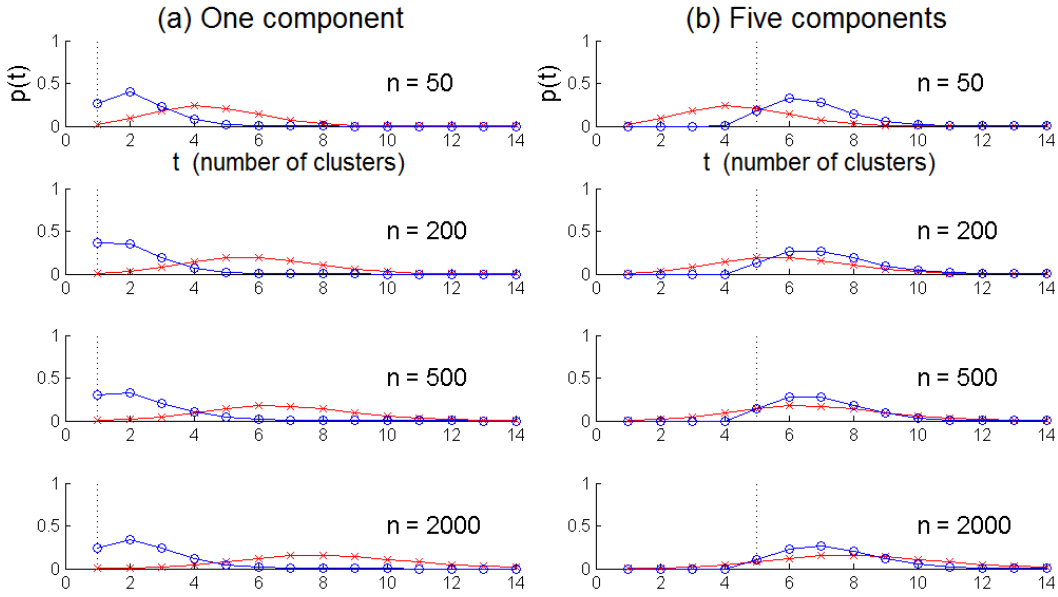

Figure 1: Prior (red x) and estimated posterior (blue o) of the number of clusters in the observed data, for a univariate normal DPM on $n$ i.i.d. samples from (a) $\mathcal{N}(0,1)$, and (b) $\sum_{k=-2}^{2} \frac{1}{5}\mathcal{N}(4k, \frac{1}{2})$. The DPM had concentration parameter $\alpha = 1$ and a Normal–Gamma base measure on the mean and precision: $\mathcal{N}(\mu \mid 0, 1/c\lambda)\mathrm{Gamma}(\lambda \mid a, b)$ with $a = 1$, $b = 0.1$, and $c = 0.001$. Estimates were made using a collapsed Gibbs sampler, with $10^4$ burn-in sweeps and $10^5$ sample sweeps; traceplots and running averages were used as convergence diagnostics. Each plot shown is an average over 5 independent runs.

simple as possible: a "standard normal DPM", that is, a DPM using univariate normal components of unit variance, with a standard normal base measure (prior on component means).

The rest of the paper is organized as follows. In Section 2, we address several pertinent questions and consider some suggestive experimental evidence. In Section 3, we formally define the DPM model under consideration. In Section 4, we give an elementary proof of inconsistency in the case of a standard normal DPM on data from one component, and in Section 5, we show that on standard normal data, a standard normal DPM is in fact severely inconsistent.

## 2   Discussion

It should be emphasized that these results do not diminish, in any way, the utility of Dirichlet process mixtures as a flexible prior on densities, i.e., for Bayesian density estimation. In addition to their widespread success in empirical studies, DPMs are backed by theoretical guarantees showing that in many cases the posterior on the density concentrates at the true density at the minimax-optimal rate, up to a logarithmic factor (see [1] and references therein).

Many researchers (e.g. [8, 9], among others) have empirically observed that the DPM posterior on the number of clusters tends to overestimate the number of components, in the sense that it tends to put its mass on a range of values greater or equal to the true number. Figure 1 illustrates this effect for univariate normals, and similar experiments with different families of component distributions yield similar results. Thus, while our theoretical results in Sections 4 and 5 (and in [7]) are asymptotic in nature, experimental evidence suggests that the issue is present even in small samples.

It is natural to think that this overestimation is due to the fact that the prior on the number of clusters diverges as $n \to \infty$, at a $\log n$ rate. However, this does not seem to be the main issue — rather, the problem is that DPMs strongly prefer having some tiny clusters and will introduce extra clusters even when they are not needed (see [7] for an intuitive explanation of why this is the case).

In fact, many researchers have observed the presence of tiny extra clusters (e.g. [8, 9]), but the reason for this has not previously been well understood, often being incorrectly attributed to the difficulty of detecting components with small weight. These tiny extra clusters are rather inconvenient, especially in clustering applications, and are often dealt with in an *ad hoc* way by simply removing them. It might be possible to consistently estimate the number of components in this way, but this remains an open question.

A more natural solution is the following: if the number of components is unknown, put a prior on the number of components. For example, draw the number of components $s$ from a probability mass function $p(s)$ on $\{1, 2, \dots\}$ with $p(s) > 0$ for all $s$, draw mixing weights $\pi = (\pi_1, \dots, \pi_s)$ (given $s$), draw component parameters $\theta_1, \dots, \theta_s$ i.i.d. (given $s$ and $\pi$) from an appropriate prior, and draw $X_1, X_2, \dots$ i.i.d. (given $s$, $\pi$, and $\theta_{1:s}$) from the resulting mixture. This approach has been widely used [10, 11, 12, 13]. Under certain conditions, the posterior on the density has been shown to concentrate at the true density at the minimax-optimal rate, up to a logarithmic factor, for any sufficiently regular true density [14]. Strictly speaking, as defined, such a model is not identifiable, but it is fairly straightforward to modify it to be identifiable by choosing one representative from each equivalence class. Subject to a modification of this sort, it can be shown (see [10]) that under very general conditions, when the data is from a finite mixture of the chosen family, such models are (a.e.) consistent for the number of components, the mixing weights, the component parameters, and the density. Also see [15] for an interesting discussion about estimating the number of components.

However, as a practical matter, when dealing with real-world data, one would not expect to find data coming exactly from a finite mixture of a known family (except, perhaps, in rare circumstances). Unfortunately, even for a model as in the preceding paragraph, the posterior on the number of components will typically be highly sensitive to misspecification, and it seems likely that in order to obtain robust estimators, the problem itself may need to be reformulated. We urge researchers interested in the number of components to be wary of this robustness issue, and to think carefully about whether they really need to estimate the number of components, or whether some other measure of heterogeneity will suffice.

## 3 Setup

In this section, we define the Dirichlet process mixture model under consideration.

### 3.1 Dirichlet process mixture model

The DPM model was introduced by Ferguson [16] and Lo [17] for the purpose of Bayesian density estimation, and was made practical through the efforts of several authors (see [18] and references therein). We will use $p(\cdot)$ to denote probabilities under the DPM model (as opposed to other probability distributions that will be considered in what follows). The core of the DPM is the so-called Chinese restaurant process (CRP), which defines a certain probability distribution on partitions. Given $n \in \{1, 2, \dots\}$ and $t \in \{1, \dots, n\}$, let $\mathcal{A}_t(n)$ denote the set of all *ordered* partitions $(A_1, \dots, A_t)$ of $\{1, \dots, n\}$ into $t$ nonempty sets. In other words,

$$\mathcal{A}_t(n) = \left\{ (A_1, \dots, A_t) : A_1, \dots, A_t \text{ are disjoint, } \bigcup_{i=1}^{t} A_i = \{1, \dots, n\}, \ |A_i| \geq 1 \ \forall i \right\}.$$

The CRP with concentration parameter $\alpha > 0$ defines a probability mass function on $\mathcal{A}(n) = \bigcup_{t=1}^{n} \mathcal{A}_t(n)$ by setting

$$p(A) = \frac{\alpha^t}{\alpha^{(n)} \, t!} \prod_{i=1}^{t} (|A_i| - 1)!$$

for $A \in \mathcal{A}_t(n)$, where $\alpha^{(n)} = \alpha(\alpha + 1) \cdots (\alpha + n - 1)$. Note that since $t$ is a function of $A$, we have $p(A) = p(A, t)$. (It is more common to see this distribution defined in terms of unordered partitions $\{A_1, \dots, A_t\}$, in which case the $t!$ does not appear in the denominator — however, for our purposes it is more convenient to use the distribution on ordered partitions $(A_1, \dots, A_t)$ obtained by uniformly permuting the parts. This does not affect the prior or posterior on $t$.)

Consider the hierarchical model

$$p(A, t) = p(A) = \frac{\alpha^t}{\alpha^{(n)}\, t!} \prod_{i=1}^{t} (|A_i| - 1)!, \tag{3.1}$$

$$p(\theta_{1:t} \mid A, t) = \prod_{i=1}^{t} \pi(\theta_i), \text{ and}$$

$$p(x_{1:n} \mid \theta_{1:t}, A, t) = \prod_{i=1}^{t} \prod_{j \in A_i} p_{\theta_i}(x_j),$$

where $\pi(\theta)$ is a prior on component parameters $\theta \in \Theta$, and $\{p_\theta : \theta \in \Theta\}$ is a parametrized family of distributions on $x \in \mathcal{X}$ for the components. Typically, $\mathcal{X} \subset \mathbb{R}^d$ and $\Theta \subset \mathbb{R}^k$ for some $d$ and $k$. Here, $x_{1:n} = (x_1, \ldots, x_n)$ with $x_i \in \mathcal{X}$, and $\theta_{1:t} = (\theta_1, \ldots, \theta_t)$ with $\theta_i \in \Theta$. This hierarchical model is referred to as a *Dirichlet process mixture (DPM) model*.

The prior on the number of clusters $t$ under this model is $p_n(t) = \sum_{A \in \mathcal{A}_t(n)} p(A, t)$. We use $T_n$ (rather than $T$) to denote the random variable representing the number of clusters, as a reminder that its distribution depends on $n$. Note that we distinguish between the terms "component" and "cluster": a *component* is part of a mixture distribution (e.g. a mixture $\sum_{i=1}^{\infty} \pi_i p_{\theta_i}$ has components $p_{\theta_1}, p_{\theta_2}, \ldots$), while a *cluster* is the set of indices of data points coming from a given component (e.g. in the DPM model above, $A_1, \ldots, A_t$ are the clusters).

Since we are concerned with the posterior distribution $p(T_n = t \mid x_{1:n})$ on the number of clusters, we will be especially interested in the marginal distribution on $(x_{1:n}, t)$, given by

$$p(x_{1:n}, T_n = t) = \sum_{A \in \mathcal{A}_t(n)} \int p(x_{1:n}, \theta_{1:t}, A, t)\, d\theta_{1:t}$$

$$= \sum_{A \in \mathcal{A}_t(n)} p(A) \prod_{i=1}^{t} \int \left( \prod_{j \in A_i} p_{\theta_i}(x_j) \right) \pi(\theta_i)\, d\theta_i$$

$$= \sum_{A \in \mathcal{A}_t(n)} p(A) \prod_{i=1}^{t} m(x_{A_i}) \tag{3.2}$$

where for any subset of indices $S \subset \{1, \ldots, n\}$, we denote $x_S = (x_j : j \in S)$ and let $m(x_S)$ denote the single-cluster marginal of $x_S$,

$$m(x_S) = \int \left( \prod_{j \in S} p_\theta(x_j) \right) \pi(\theta)\, d\theta. \tag{3.3}$$

### 3.2 Specialization to the standard normal case

In this note, for brevity and clarity, we focus on the univariate normal case with unit variance, with a standard normal prior on means — that is, for $x \in \mathbb{R}$ and $\theta \in \mathbb{R}$,

$$p_\theta(x) = \mathcal{N}(x \mid \theta, 1) = \frac{1}{\sqrt{2\pi}} \exp(-\tfrac{1}{2}(x - \theta)^2), \quad \text{and}$$

$$\pi(\theta) = \mathcal{N}(\theta \mid 0, 1) = \frac{1}{\sqrt{2\pi}} \exp(-\tfrac{1}{2}\theta^2).$$

It is a straightforward calculation to show that the single-cluster marginal is then

$$m(x_{1:n}) = \frac{1}{\sqrt{n+1}}\, p_0(x_{1:n}) \exp \left( \frac{1}{2} \frac{1}{n+1} \left( \sum_{j=1}^{n} x_j \right)^2 \right), \tag{3.4}$$

where $p_0(x_{1:n}) = p_0(x_1) \cdots p_0(x_n)$ (and $p_0$ is the $\mathcal{N}(0, 1)$ density). When $p_\theta(x)$ and $\pi(\theta)$ are as above, we refer to the resulting DPM as a *standard normal DPM*.

## 4 Simple example of inconsistency

In this section, we prove the following result, exhibiting a simple example in which a DPM is inconsistent for the number of components: even when the true number of components is 1 (e.g. $\mathcal{N}(\mu, 1)$ data), the posterior probability of $T_n = 1$ does not converge to 1. Interestingly, the result applies even when $X_1, X_2, \dots$ are identically equal to a constant $c \in \mathbb{R}$. To keep it simple, we set $\alpha = 1$; for more general results, see [7].

**Theorem 4.1.** *If $X_1, X_2, \dots \in \mathbb{R}$ are i.i.d. from any distribution with $\mathbb{E}|X_i| < \infty$, then with probability 1, under the standard normal DPM with $\alpha = 1$ as defined above, $p(T_n = 1 \mid X_{1:n})$ does not converge to 1 as $n \to \infty$.*

*Proof.* Let $n \in \{2, 3, \dots\}$. Let $x_1, \dots, x_n \in \mathbb{R}$, $A \in \mathcal{A}_2(n)$, and $a_i = |A_i|$ for $i = 1, 2$. Define $s_n = \sum_{j=1}^n x_j$ and $s_{A_i} = \sum_{j \in A_i} x_j$ for $i = 1, 2$. Using Equation 3.4 and noting that $1/(n+1) \leq 1/(n+2) + 1/n^2$, we have

$$\sqrt{n+1} \, \frac{m(x_{1:n})}{p_0(x_{1:n})} = \exp\left(\frac{1}{2} \frac{s_n^2}{n+1}\right) \leq \exp\left(\frac{1}{2} \frac{s_n^2}{n+2}\right) \exp\left(\frac{1}{2} \frac{s_n^2}{n^2}\right).$$

The second factor equals $\exp(\frac{1}{2}\overline{x}_n^2)$, where $\overline{x}_n = \frac{1}{n}\sum_{j=1}^n x_j$. By the convexity of $x \mapsto x^2$,

$$\left(\frac{s_n}{n+2}\right)^2 \leq \frac{a_1 + 1}{n+2}\left(\frac{s_{A_1}}{a_1 + 1}\right)^2 + \frac{a_2 + 1}{n+2}\left(\frac{s_{A_2}}{a_2 + 1}\right)^2,$$

and thus, the first factor is less or equal to

$$\exp\left(\frac{1}{2} \frac{s_{A_1}^2}{a_1 + 1} + \frac{1}{2} \frac{s_{A_2}^2}{a_2 + 1}\right) = \sqrt{a_1 + 1}\sqrt{a_2 + 1} \, \frac{m(x_{A_1}) \, m(x_{A_2})}{p_0(x_{1:n})}.$$

Hence,

$$\frac{m(x_{1:n})}{m(x_{A_1}) \, m(x_{A_2})} \leq \frac{\sqrt{a_1 + 1}\sqrt{a_2 + 1}}{\sqrt{n+1}} \exp(\tfrac{1}{2}\overline{x}_n^2). \tag{4.1}$$

Consequently, we have

$$\frac{p(x_{1:n}, T_n = 2)}{p(x_{1:n}, T_n = 1)} \overset{\text{(a)}}{=} \sum_{A \in \mathcal{A}_2(n)} n \, p(A) \, \frac{m(x_{A_1}) \, m(x_{A_2})}{m(x_{1:n})}$$

$$\overset{\text{(b)}}{\geq} \sum_{A \in \mathcal{A}_2(n)} n \, p(A) \, \frac{\sqrt{n+1}}{\sqrt{|A_1| + 1}\sqrt{|A_2| + 1}} \exp(-\tfrac{1}{2}\overline{x}_n^2)$$

$$\overset{\text{(c)}}{\geq} \sum_{\substack{A \in \mathcal{A}_2(n): \\ |A_1| = 1}} n \, \frac{(n-2)!}{n! \, 2!} \, \frac{\sqrt{n+1}}{\sqrt{2}\sqrt{n}} \exp(-\tfrac{1}{2}\overline{x}_n^2)$$

$$\overset{\text{(d)}}{\geq} \frac{1}{2\sqrt{2}} \exp(-\tfrac{1}{2}\overline{x}_n^2),$$

where step (a) follows from applying Equation 3.2 to both numerator and denominator, plus using Equation 3.1 (with $\alpha = 1$) to see that $p(A) = 1/n$ when $A = (\{1, \dots, n\})$, step (b) follows from Equation 4.1 above, step (c) follows since all the terms in the sum are nonnegative and $p(A) = (n-2)!/n! \, 2!$ when $|A_1| = 1$ (by Equation 3.1, with $\alpha = 1$), and step (d) follows since there are $n$ partitions $A \in \mathcal{A}_2(n)$ such that $|A_1| = 1$.

If $X_1, X_2, \dots \in \mathbb{R}$ are i.i.d. with $\mu = \mathbb{E}X_j$ finite, then by the law of large numbers, $\overline{X}_n = \frac{1}{n}\sum_{j=1}^n X_j \to \mu$ almost surely as $n \to \infty$. Therefore,

$$p(T_n = 1 \mid X_{1:n}) = \frac{p(X_{1:n}, T_n = 1)}{\sum_{t=1}^\infty p(X_{1:n}, T_n = t)} \leq \frac{p(X_{1:n}, T_n = 1)}{p(X_{1:n}, T_n = 1) + p(X_{1:n}, T_n = 2)}$$

$$\leq \frac{1}{1 + \frac{1}{2\sqrt{2}}\exp(-\tfrac{1}{2}\overline{X}_n^2)} \overset{\text{a.s.}}{\longrightarrow} \frac{1}{1 + \frac{1}{2\sqrt{2}}\exp(-\tfrac{1}{2}\mu^2)} < 1.$$

Hence, almost surely, $p(T_n = 1 \mid X_{1:n})$ does not converge to 1. $\qquad\square$

# 5 Severe inconsistency

In the previous section, we showed that $p(T_n = 1 \mid X_{1:n})$ does not converge to 1 for a standard normal DPM on any data with finite mean. In this section, we prove that in fact, it converges to 0, at least on standard normal data. This vividly illustrates that improperly using DPMs in this way can lead to entirely misleading results. The key step in the proof is an application of Hoeffding's strong law of large numbers for U-statistics.

**Theorem 5.1.** *If $X_1, X_2, \ldots \sim \mathcal{N}(0,1)$ i.i.d. then*

$$p(T_n = 1 \mid X_{1:n}) \xrightarrow{\text{Pr}} 0 \quad \text{as } n \to \infty$$

*under the standard normal DPM with concentration parameter $\alpha = 1$.*

*Proof.* For $t = 1$ and $t = 2$ define

$$R_t(X_{1:n}) = n^{3/2} \frac{p(X_{1:n}, T_n = t)}{p_0(X_{1:n})}.$$

Our method of proof is as follows. We will show that

$$R_2(X_{1:n}) \xrightarrow[n \to \infty]{\text{Pr}} \infty$$

(or in other words, for any $B > 0$ we have $\mathbb{P}(R_2(X_{1:n}) > B) \to 1$ as $n \to \infty$), and we will show that $R_1(X_{1:n})$ is bounded in probability:

$$R_1(X_{1:n}) = O_P(1)$$

(or in other words, for any $\varepsilon > 0$ there exists $B_\varepsilon > 0$ such that $\mathbb{P}(R_1(X_{1:n}) > B_\varepsilon) \leq \varepsilon$ for all $n \in \{1, 2, \ldots\}$). Putting these two together, we will have

$$p(T_n = 1 \mid X_{1:n}) = \frac{p(X_{1:n}, T_n = 1)}{\sum_{t=1}^{\infty} p(X_{1:n}, T_n = t)} \leq \frac{p(X_{1:n}, T_n = 1)}{p(X_{1:n}, T_n = 2)} = \frac{R_1(X_{1:n})}{R_2(X_{1:n})} \xrightarrow[n \to \infty]{\text{Pr}} 0.$$

First, let's show that $R_2(X_{1:n}) \to \infty$ in probability. For $S \subset \{1, \ldots, n\}$ with $|S| \geq 1$, define $h(x_S)$ by

$$h(x_S) = \frac{m(x_S)}{p_0(x_S)} = \frac{1}{\sqrt{|S|+1}} \exp\left(\frac{1}{2} \frac{1}{|S|+1}\left(\sum_{j \in S} x_j\right)^2\right),$$

where $m$ is the single-cluster marginal as in Equations 3.3 and 3.4. Note that when $1 \leq |S| \leq n-1$, we have $\sqrt{n}\, h(x_S) \geq 1$. Note also that $\mathbb{E}h(X_S) = 1$ since

$$\mathbb{E}h(X_S) = \int h(x_S)\, p_0(x_S)\, dx_S = \int m(x_S)\, dx_S = 1,$$

using the fact that $m(x_S)$ is a density with respect to Lebesgue measure. For $k \in \{1, \ldots, n\}$, define the U-statistics

$$U_k(X_{1:n}) = \frac{1}{\binom{n}{k}} \sum_{|S|=k} h(X_S)$$

where the sum is over all $S \subset \{1, \ldots, n\}$ such that $|S| = k$. By Hoeffding's strong law of large numbers for U-statistics [19],

$$U_k(X_{1:n}) \xrightarrow[n \to \infty]{\text{a.s.}} \mathbb{E}h(X_{1:k}) = 1$$

for any $k \in \{1, 2, \dots\}$. Therefore, using Equations 3.1 and 3.2 we have that for any $K \in \{1, 2, \dots\}$ and any $n > K$,

$$
\begin{aligned}
R_2(X_{1:n}) &= n^{3/2} \sum_{A \in \mathcal{A}_2(n)} p(A) \frac{m(X_{A_1}) \, m(X_{A_2})}{p_0(X_{1:n})} \\
&= n \sum_{A \in \mathcal{A}_2(n)} p(A) \sqrt{n} \, h(X_{A_1}) \, h(X_{A_2}) \\
&\geq n \sum_{A \in \mathcal{A}_2(n)} p(A) \, h(X_{A_1}) \\
&= n \sum_{k=1}^{n-1} \sum_{|S|=k} \frac{(k-1)! \, (n-k-1)!}{n! \, 2!} \, h(X_S) \\
&= \sum_{k=1}^{n-1} \frac{n}{2k(n-k)} \frac{1}{\binom{n}{k}} \sum_{|S|=k} h(X_S) \\
&= \sum_{k=1}^{n-1} \frac{n}{2k(n-k)} \, U_k(X_{1:n}) \\
&\geq \sum_{k=1}^{K} \frac{n}{2k(n-k)} \, U_k(X_{1:n}) \\
&\xrightarrow[n \to \infty]{\text{a.s.}} \sum_{k=1}^{K} \frac{1}{2k} = \frac{H_K}{2} > \frac{\log K}{2}
\end{aligned}
$$

where $H_K$ is the $K^{\text{th}}$ harmonic number, and the last inequality follows from the standard bounds [20] on harmonic numbers: $\log K < H_K \leq \log K + 1$. Hence, for any $K$,

$$
\liminf_{n \to \infty} R_2(X_{1:n}) > \frac{\log K}{2} \qquad \text{almost surely,}
$$

and it follows easily that

$$
R_2(X_{1:n}) \xrightarrow[n \to \infty]{\text{a.s.}} \infty.
$$

Convergence in probability is implied by almost sure convergence.

Now, let's show that $R_1(X_{1:n}) = O_P(1)$. By Equations 3.1, 3.2, and 3.4, we have

$$
\begin{aligned}
R_1(X_{1:n}) &= n^{3/2} \frac{p(X_{1:n}, T_n = 1)}{p_0(X_{1:n})} = \sqrt{n} \, \frac{m(X_{1:n})}{p_0(X_{1:n})} \\
&= \frac{\sqrt{n}}{\sqrt{n+1}} \exp\left( \frac{1}{2} \frac{n}{n+1} \left( \frac{1}{\sqrt{n}} \sum_{i=1}^{n} X_i \right)^2 \right) \leq \exp(Z_n^2/2)
\end{aligned}
$$

where $Z_n = (1/\sqrt{n}) \sum_{i=1}^{n} X_i \sim \mathcal{N}(0, 1)$ for each $n \in \{1, 2, \dots\}$. Since $Z_n = O_P(1)$ then we conclude that $R_1(X_{1:n}) = O_P(1)$. This completes the proof. $\square$

### Acknowledgments

We would like to thank Stu Geman for raising this question, and the anonymous referees for several helpful suggestions that improved the quality of this manuscript. This research was supported in part by the National Science Foundation under grant DMS-1007593 and the Defense Advanced Research Projects Agency under contract FA8650-11-1-715.

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
