[Reviews · NeurIPS 2013]

Submitted by Assigned_Reviewer_4

This paper addresses one simple but potentially very important point: That Dirichlet process mixture models can be inconsistent in the number of mixture components that they infer. This is important because DPs are nowadays widely used in various types of statistical modeling, for example when building clustering type algorithms. This can have real-world implications, for example when clustering breast cancer data with the aim of identifying distinct disease subtypes. Such subtypes are used in clinical practice to inform treatment, so identifying the correct number of clusters (and hence subtypes) has a very important real-world impact.

The paper focuses on proofs concerning two specific cases where the DP turns out to be inconsistent. Both consider the case of the "standard normal DPM", where the likelihood is a univariate normal distribution with unit variance, the mean of which is subject to a normal prior with unit variance. The first proof shows that, if the data are drawn i.i.d. from a zero-mean, unit-variance normal (hence matching the assumed DPM model), P(T=1 | data) does not converge to 1. The second proof takes this further, demonstrating that in fact P(T=1 | data) -> 0

This seems to me to be something of which the DPM community needs to be aware and to understand the nature of. To that end, I would very much like to see the following additions to the discussion in the paper.

1 - Why does this happen? What is it about the DPM that causes it to be inconsistent here?

2 - Can it be shown whether or not this is also potentially a problem when the are 2+ groups of data points in the data?

3 - Can the author/s make some suggestions for ways to fix this? There is some discussion of this in the introduction, but as someone who uses DPMs a lot, I'd find it very interesting to have some practical advice in the discussion.
Summary: This paper makes an interesting and potentially important point about Dirichlet process mixture models.

Submitted by Assigned_Reviewer_5

The title of this paper is much like the paper itself: to-the-point, descriptive, and readable. "A simple example of Dirichlet process mixture inconsistency for the number of components" delivers on its promise by providing two easy-to-understand demonstrations of the severity of the problem of using Dirichlet process mixtures to estimate the number of components in a mixture model. The authors start by demonstrating that making such a component-cardinality estimate is widespread in the literature (and therefore a problem deserving of interest), briefly describe the Dirichlet process mixture (DPM) model (with particular emphasis on the popular normal likelihood case), and then demonstrate with a simple single-component mixture example how poorly estimation of component cardinality can go (their convincing answer: very poorly).

Not only was the paper enjoyable to read but, refreshingly, didn't try to fit 20 pages of material into an 8 page limit.

One potential criticism of this paper is that this result should be well-known in some sense in the community. It is a widely cited result that the number of components of a DPM grows logarithmically with the size of the data (and indeed the popularity of the Pitman-Yor process extension of the DPM is due to the different way in which the number of components grows: as an almost sure power of the size of the data rather than merely logarithmically). However, this paper is still necessary for two key reasons. (1) Existing results deal only obliquely with consistency in component cardinality, and this paper seems to be the first to prove component-cardinality inconsistency directly. (2) The authors make a case that the community, in spite of existing results, uses the DPM to estimate the ``true'' number of components in a data set. Since there are cases where the number of components in a mixture can indeed be expected to grow with the size of the data (cf. the HDP extension to LDA for topic modeling), the authors could make an even more compelling argument by highlighting in text and via their citations those cases in the literature where other authors use the DPM while assuming a fixed (i.e., not varying with the number of data points) number of clusters in a data set of any size. The authors might, in particular, make the distinction between the number of components in the DPM generative model (infinite, as on line 38, page 1) and the number of components "active" in a given data set. It is not clear to me that using the DPM to estimate the active number of components for a data set of size N, while assuming that a data set of greater size would exhibit more components, is immediately problematic; the real problem seems to be estimating a fixed, underlying number of components. Some care in describing this subtlety might be useful.

Perhaps my one major concern is that at a number of times in the paper the authors make what appear to be unsupported claims by referring to seemingly already-completed future work---that is, work that is not in the current paper or cited. E.g., on line 46, page 1, "as we will show elsewhere" and line 66, page 2 "Further details will be provided elsewhere", and even the extensions to general alpha in Proposition 3.1 and Theorem 4.1. It seems inappropriate to state results that are not proved in the current paper or elsewhere in existing published literature---except perhaps as proposed "future work" in the conclusion. There is no way to judge the merits of these results until they are given in full. Moreover, I think the paper stands on its own merits, and these further results are not strictly necessary (though they might be nice bonuses). Here are some more details:
* For page 1, line 46, I suggest removing the sentence or giving the full result; the counterexamples provided either stand on their own or they don't.
* Page 2, lines 64-66 make some interesting claims but again do not seem to be supported by any citations or work in the current paper. One option is to simply add the sampler (and some timing and experimental results) to this paper. Another is to delete the sentences or move them to a "future work" discussion. I agree that it's useful to point out that there are practical alternatives to the DPM. But part of what's problematic about these unsupported claims is that the reader can't tell whether many types of inference (including approximations that are not MCMC) are available or evaluate how the two MCMC samplers compare in speed, ease of use, etc.
* In Proposition 3.1 and Theorem 4.1, I suggest providing a full proof of the general alpha case in the supplementary material (or in the main text if it is indeed "trivial" as stated on page 4, line 167).

I also have a number of minor comments:
* Page 2, line 72: Perhaps say a bit more explicitly about why the DPM cannot be used as a heuristic. I assume the idea here is that, given a fixed number of underlying components, we expect the same amount of heterogeneity across a large range of data cardinalities but can also expect the DPM to return very different estimates of heterogeneity as data-size changes in these cases.
* Page 2, line 101: Introduce alpha in the text as a hyperparameter of the model before using it. Notation for the CRP is inconsistent across different disciplines, so it would help to name alpha as the "concentration parameter" and give its allowable range of values.
* Page 2: This is a nice, careful treatment of ordered vs. unordered partitions.
* Page 3, line 126: The "component" vs "cluster" distinction is somewhat unclear. Consider drawing upon notation to clarify.
* Page 4, line 107: Note that X_1, X_2, ... ~ N(0,1) is a different assumption than in the statement of Proposition 3.1. I believe the proof given here holds for the more general case of the Proposition 3.1 statement though. Interestingly, it seems to hold when X_1, X_2, etc are iid from a single point mass. If indeed true, that seems like a compelling special case that may be worth mentioning.
* Page 6, line 304: Consider making this remark about p(A) = 1/n in this special case the first time it is used instead (on page 4, line 192).

========
Update after rebuttal:
I have read the author response, which addressed my few and minor concerns. I look forward to reading the final product.
Summary: This paper makes a compelling argument about our community's sometimes incorrect use of the Dirichlet process mixture model to estimate the number of components of a mixture. The authors present succinct, highly readable theoretical results to demonstrate their claims.

Submitted by Assigned_Reviewer_6

This paper brings up an excellent point: consistency in density doesn't imply consistency in parameters, and they demonstrate that the Dirichlet Process Mixture is in fact inconsistent in the parameters (specifically, in "K" -- the effective number of clusters). The paper demonstrates that the DPM will not converge to K=1 when the data given are from a single Gaussian. They give two proofs, one simpler proof saying that P(K=1) does not go to 1 as N->inf, and one giving that P(K=1)->0.

This work supports the commonly held belief that DPMs will often find too many clusters.

Summary: This paper shows that Dirichlet Process Mixtures are inconsistent in the number of effective clusters, K.
Author Feedback

Author rebuttal: We would like to thank the reviewers for generously spending their time and effort in reviewing our paper. Several helpful comments were made; we will briefly address the main points.

Reviewer_5 and Reviewer_6 both expressed dissatisfaction with the places where we mentioned additional results that are not covered in this paper. This is a valid criticism which we will address by either removing or proving these particular claims.

Reviewer_4 posed three important questions. As it turns out, these are the subject of work that we will be publishing in the near future. Thus, perhaps these questions would be most appropriately addressed in a "future work" section.

In Proposition 3.1, Reviewer_5 and Reviewer_6 noticed that we accidentally used a version of the proof for the case of standard normal data, instead of arbitrary data with a finite mean. At the end of the proof, the following two substitutions should be made:

"If X_1,X_2,... ~ N(0,1) i.i.d. then by the law of large numbers, \bar{X}_n = (1/n)\sum X_j -> 0"

should instead be

"If X_1,X_2,... are i.i.d. with \mu = E(X_j) finite, then by the law of large numbers, \bar{X}_n = (1/n)\sum X_j -> \mu"

and the limit

1/(1 + 1/(2 \sqrt{2}))

should instead be

1/(1 + exp(-\mu^2/2)/(2 \sqrt{2})).

Reviewer_5 provided a very thorough and insightful review (which we greatly appreciate), and made a number of other good suggestions which we intend to address in the revised manuscript.

Thank you, again, for your time and consideration.